# The Nexus among Competitively Valued Exchange Rates, Price Level, and Growth Performance in the Turkish Economy; New Insight from the Global Value Chains

**Umar Aliyu Shuabiu [1], Mohammed A. M. Usman [1] and Behiye Çavuşoğlu [2,*]**

1   Department of Economics, Near East University, Boulevard, Nicosia 99138, Cyprus;
    umar.aliyu@neu.edu.tr (U.A.S.); amkmgella@gmail.com (M.A.M.U.)
2   Department of Innovation and Knowledge Management, Near East University, Boulevard,
    Nicosia 99138, Cyprus
*   Correspondence: behiye.cavusoglu@neu.edu.tr

**Abstract:** Currently, global value chains (GVCs) are increasingly shaping the global economy, covering a growing share of international trade, GDP, and employment globally. Global trade is impacted by the emergence of GVCs in areas as diverse as commodities, electronics, and business service outsourcing, among other areas, since the countries involved in the GVCs hold some value(s) and benefit(s) from the exports of the finished product. In this study, the nexus among Competitively Valued Exchange Rates, Price level, and Growth Performance in the Turkish Economy; New insight from the GVCs is investigated using annual data from 1980 to 2020 within the framework of the ARDL bound test, Bayer and Hanck Cointegration (BHC) test, and ECM. The study results revealed that the relationship among real effective exchange rate, exports, and imports induced economic performance and external trade competitiveness particularly when directed at GVCs in both the short and long run. The study recommends that policies enhancing a 10% equilibrium convergence are required annually to competitively minimize the dependence on foreign value-added inputs by importing only world-class inputs for value addition and exports benefits in the competitive GVCs world. Furthermore, monetary policy, GVCs, and economic growth should be investigated.

**Keywords:** Turkey; economic growth; ARDL bounds test; exchange rate; foreign trade

**JEL Classification:** F: International economics; F4: macroeconomic aspects of international trade and finance; F31—Foreign Exchange

## 1. Introduction

The Turkish economy is a promising and emerging open market economy (i.e., having 21.97% of its exports and 24.85% of imports). It is one of the leading agricultural producers, along with construction and building materials, end users electronics, house, and office appliances, among others. In the early 1970s, the Turkish economy was relatively stable because increased investment and government spending boosted the growth process. However, the lack of tough structural reforms from the late 1970s onward caused economic growth to slow down to about 1.5 and 1.3% in 1979 and 1980, respectively. Consequently, Turkey was confronted by an unprecedented balance of payments (i.e., current account) deficit generally financed by external obligations, which increased the foreign debt to about 16.2 billion USD. Furthermore, inflation reached triple-digit levels, debt servicing cost 33% of exports of goods and services, and unemployment reached approximately 15% by 1980. Although gold acted as a hedge against currency fluctuations and inflation movements for a long time for the United States and Turkey, it failed to offer protection against currency fluctuations and inflation movements in the period of hyperinflation in the Turkish economy. Instead, it acted against risk reduction concerning only a currency investment and portfolio risk management about return accumulation (Sui et al. 2020).

Nevertheless, the Turkish economy began to embark on new adjustment strategies with a greater focus on export expansion and a market forces mechanism to help finance imports because of the sizeable volumes of new lending and debt relief the country received from its creditors between 1980 and 1984. The Turkish lira was devalued and a flexible exchange rate was adopted, among other reforms. There was a remarkable improvement in the economic performance in terms of merchandise exports, which rose to US$8.3 in 1985 billion from US$2.3 in 1979, but inflation decreased to double-digit levels. In the same period, merchandise imports increased to US$11.2 in 1985 billion from US$4.8 in 1979. This boosted the Turkish economy's creditworthiness in the international market and consequently increased the economy's global competitiveness. International openness, oil prices, Real GDP per capita relative to trading partners, and fiscal expenditures are the main fundamental determinants of Turkish real effective exchange rates because a constant rise in GDP per capita relative to major trading partners' measure is associated with the appreciation pressure on equilibrium exchange rates (Vural and Burçak 2019). In 1994, Turkey's foreign debt was rated at below investment grade by international credit agencies. This caused a large–scale capital outflow and a decline in the value of exchange rates, and consequently, the Turkish lira declined by 76% against the United States dollar.

Doubts remain in terms of the sustainability and stability of the future growth trajectory in Turkey because the country has been excessively reliant on foreign credit, which comes at the cost of highly volatile real GDP growth rates. Consequently, Turkey is considered to be one of the most credit-dependent nations across all emerging markets since its current account deficit is financed mainly by short-term debt-creating inflows (Özer and Malovic 2020). Additionally, real GDP declined by 5% after the improvement in 1992 and 1993, while exports and imports increased and inflation reached triple-digits. Turkey's currency crises during the period in question were coupled with the global liquidity conditions, banking sector weaknesses, fiscal imbalances, as well as capital outflow (Feridun 2008). Accordingly, an increase in geopolitical risks reduces tourist arrivals within the short run in Turkey, but a decrease in the Geopolitical Risk Index has no effect in the short run, and there is evidence of an asymmetric effect in the long run (Demir et al. 2020). However, Turkey's severe debt crisis brought about a series of negotiations and debt rescheduling agreements with its creditors because the debt issues were considered necessary in order to resolve the issues and to implement the necessary economic reforms (Onder 1990).

The political stability, government effectiveness, and the rule of law in the host nation can attract foreign direct investment (Heavilin and Hilmi 2020). Contrastingly, Turkey's performance since 2000 has been impressive because it has also recovered well from the global crisis of 2008/09 to the extent that, as an emerging economy, it has represented an attractive marketplace for investments for almost two decades. It achieved strong economic growth because of the positive economic and banking reforms it applied from 2002 to 2007. Although inflation declined in the 2000s, it increased from 8.9% in 2010 to 10.5% in 2011. As a result of the recent political crisis and attempted coup in July 2016, growth in Turkey dropped from 6.1% in 2015 to 2.1% in 2016 (World Bank Annual Report 2016). In addition, annual tourism in Turkey also dropped by 36% because business and consumer confidence were both affected, and this has applied increased pressure on the Turkish lira and restricted the growth of local credit based on the decline in the interest rate levels (World Bank Annual Report 2016). Since 2013, an extensive debate has been ongoing concerning the exchange rate in Turkey and the stronger Turkish lira in exchange for major international currencies, particularly the US dollar. GDP is a significant variable that influences imports and exports in an economy (Fidan 2006). The value of imports and exports, in the long run, determines the fluctuations in the demand and supply of the currency. The inflation level also affects the level of the exchange rate in the economy since traders look for the worthiness of a currency to determine optimal investments that would provide desired returns (Zamir et al. 2017).

The contribution of this study stems from a theoretical framework in which the Mundell–Fleming model was employed to guide the investigation. The model describes

the impossible trinity of maintaining a fixed exchange rate, free capital mobility, and independent monetary policy. Therefore, two of the three can be achieved simultaneously, particularly in a small open economy, because the model argues that an economy is not big enough to influence foreign incomes and global level of interest rates. This contribution is seldom considered in the previous studies, and this study seeks to fill this gap as well. However, the nexus among competitively valued exchange rates, price levels, and growth performance in the Turkish economy, particularly when directed at participation in the global value chains (GVCs) has not been investigated in the literature related to the Turkish economy to the best of the authors' knowledge. For example, this study incorporated the recent Bayer and Hanck cointegration test to further justify the robustness of the empirical relationship among the considered variables of interest. This is seldom considered by previous studies. Although, there may be growth in value addition in the imports and exports due to an increase in productivity and the growth consequence of linkages. This study seeks to fill this gap in the literature. Second, the study provides hints for spillover effects of how the value of exchange rate in the Turkish economy is influenced by the level of merchandise trade and geopolitical risk, which includes the country's imports and exports to help the Turkish economy to actively participate in the GVCs. Finally, the study opens ways for the research and analysis of GVCs and economic performance status of the Turkish economy to be known and predicted and help the relevant authorities and institutions to regulate and design effective policies affecting its competitive participation in the GVCs since the importance of an effective exchange rate system has been recognized as an essential global developmental goal and can be helpful in determining competitive participation in the GVCs world. A weak or strong currency can lead to a nation's trade deficit or surplus overtime, respectively.

The paper is organized into five sections: Section 2 presents a literature review and theoretical framework of previous studies to serve as the basis for this study, Section 3 is the methodology of the paper, and Section 4 presents the empirical results, while the last section presents the conclusion of the study.

## 2. Literature Review

Ergin and Filiz (2017) examined the association between terms of trade and exchange rate on the gross domestic product in Turkey using the autoregressive distributed lag bounds test approach with monthly time series data from 2005:01 and 2015:06 to evaluate whether fluctuations in exchange rates and terms of trade have any effect on Turkey's GDP. The industrial production index was used as a proxy for economic growth, terms of trade as the ratio of imports as well as exports, and the exchange rate. The findings from the study revealed that changes in terms of trade and exchange rates have positive effects on the industrial production index of Turkey. By implication, fluctuations in exchange rates and terms of trade positively affect the economic growth of Turkey. However, the Turkish economy is firmly incorporated with a volatility spillover from international markets because fluctuations in global or domestic markets regardless of borders instantly extend to further domestic markets (Alkan and Çiçek 2020). For instance, geopolitical risks stemming from Saudi Arabia and Russia affect Turkish financial stability differently since Saudi Arabia is geopolitically dependent on Turkey, while Russia is an economic and political competitor. For example, the foreign exchange market has become more vulnerable and exposed to future financial crises. Accordingly, higher geopolitical risk in Russia results in higher financial stability in Turkey, while the higher geopolitical risk in Saudi Arabia results in higher economic vulnerability in Turkey (Mansour-Ichrakieh and Zeaiter 2019). Financing constraints have a negative effect on employment generation in the Turkish economy. For example, maturing long-term debt decreases employment growth in small firms, particularly in periods of financial turmoil. However, there is no clear evidence of its effects on large firms (Demirhan and Aldan 2020). Domestic credit to the private sector advancement has a positive impact on growth performance in an economy (Shuaibu et al. 2018). The import dependency of exports increases, but is generally lower

and stable for production over time, and this difference is credited mainly to the services sector due to its low import dependency rather than a significant share of output.

Using autoregressive vectors to evaluate the strengths and weaknesses of the imports and exports of agricultural goods given the exchange rate levels, Fidan (2006) studied the impact of real effective exchange rate on Turkish agricultural trade from 1970–2004. The variables employed included the real effective exchange rate, consumer price index, import and export indexes, and the national gross product index. The study used impulse response, the Granger causality test, and the cointegration procedure to analyze the relationship among the variables of interest. The Granger causality test revealed that the exchange rate affects agricultural exports and imports in Turkey. Another finding from the research was that the exchange rate has a more significant effect on exports and imports in the long run than in the short run. Large inflows of Exchange-Traded Funds increase exchange rate volatility for simultaneous and lagged effect models, while large outflows of Exchange-Traded Funds are followed by exchange rate depreciation with less exchange rate uncertainty (Sakarya and Ekinci 2020). In contrast, investigated the impact of foreign trade on GDP in Turkey between 1975 and 2014 using a Granger causality approach. The findings of the study showed that there was no unidirectional or bidirectional causality between foreign trade and GDP in Turkey. Similarly, in another study, baye examined the short and long-term effects of the foreign exchange rate, gross domestic product, political rights, and overseas income on the GDP for Turkey from 1980 to 2010. They employed the cointegration technique and vector error correction mechanism (VECM). The results from the study revealed that exchange rates have no effect on foreign trade in Turkey. Bahmani-Oskooee et al. (2017) employed a non-linear ARDL approach. They found that both appreciation and depreciation of the exchange rate had considerable short-run effects on domestic production. Another finding from the research revealed that both appreciation and depreciation of the lira are expansionary. Studies that have attempted to answer the question regarding whether devaluation is expansionary or contractionary include the works of Ratha (2010) as well as Narayan and Narayan (2007). Accordingly, Sweidan (2013) found that a decline in Jordan's competitiveness and exchange rate only had a significant impact on the imports and exports of the Jordanian economy in the short run and opposed the adoption of the "policy of devaluation", because they argued that instability would prevail in the foreign exchange market in Jordan. Monetary and fiscal policy shocks are complementary in response to aggregate demand and aggregate supply shocks or when interest rate, tax revenue, and government spending shocks hit the economy (Büyükbaşaran et al. 2020). Money supply and interest rates, USD-Turkish lira exchange rates, and real effective exchange rates have substantial predictive power for stock price fluctuations at different frequencies. This indicates the need for innovative policy patterns that can control risk in the financial market in Turkey (Kassouri and Altıntaş 2020). Others that have drawn attention to the impact of exchange rate and other macroeconomic indicators such as prices and trade balance on output using different methodologies include (Kandil 2004; Kandil and Mirzaie 2005; Tadesse 2009; Bahmani-Oskooee and Kandil 2007; Nabli and Véganzonès-Varoudakis 2004). Similarly, Joof and Jallow (2020) investigated the effects of interest rate and inflation on the exchange rate in the Gambia from 2007M1–2018M12, using the "dynamic model of Fully Modified Ordinary Least Square (FMOLS), Dynamic Ordinary Least Square (DOLS), and Canonical Cointegrating Regression (CCR)." The long-run findings showed a positive relationship between inflation and exchange rate, suggesting that a surging inflation rate will cause the Gambian dalasi to appreciate against the US dollar.

Studies related to the nexus among competitively valued exchange rates, price level, and growth performance in Turkey with new insights from the global value chain have produced mixed results, and there has been no widespread consensus. This is due to the use of different methodologies, variables, proxies, and sample countries.

However, the preceding trends in the exchange rate, imports, and exports in the previous studies have not covered the GVCs process. They have not investigated the

nexus among competitively valued exchange rates, price levels, and growth performance in the Turkish economy, mainly when directed at participation in the global value chains to the best of the authors' knowledge. However, there may be growth in value addition in the imports and exports due to increased productivity and the growth consequence of linkages. This study seeks to fill this gap in the literature since the importance of an effective exchange rate system has been recognized as a fundamental global developmental goal and can help determine competitive participation in the GVCs world. A weak or strong currency can lead to a nation's trade deficit or surplus over time.

The mechanisms and channels of valued exchange rates and minimal price level will have a significant effect on growth performance, as depicted in Figure 1.

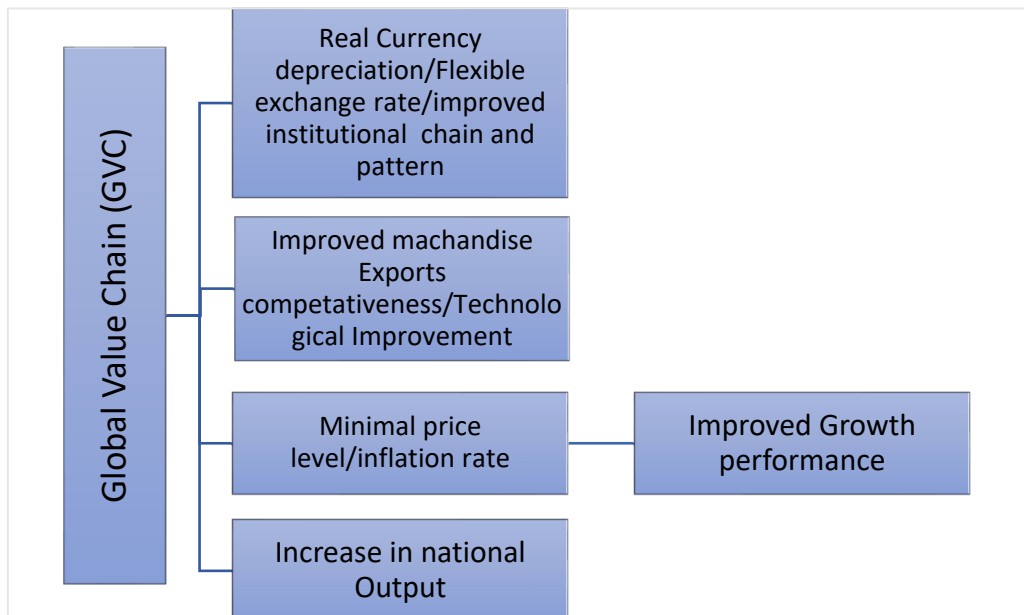

**Figure 1.** Schematic Framework for Valued exchange rate, Price level, and Growth Performance Relationship. Source: Authors.

From the schematic in Figure 1, the structure and dynamics of various actors involved in a given industry, the global value chains (GVCs) system, which helps one to understand how global industries are structured and analyzed. The GVCs technique is a valuable tool for tracing the evolving trends of global production in today's globalized economy with very complex industry interactions, connecting geographically distributed activities and actors of a single industry, and assessing the roles they play in both developed and developing countries. The structure of the GVCs focuses on the value-added sequences within a sector, from conception to output and final use. It discusses job descriptions, technology, requirements, legislation, products, procedures, and markets, in particular industries and locations, offering a holistic view of top-down and bottom-up global industries (Gereffi 1994) and (Globalvaluechains.org 2011).

Nowadays, the internet has improved access to both market and global information and inevitably induces procurement processes the world over. (PAP 2021). For example, the internet of things offers the possibility of an immediate and advanced interface between an object and its environment (Witkowski 2017). Thus, the internet of things enhances the distribution of goods by predicting failure and routinely providing other explanations to improve the supply chain using system tracking facilities (Dobrowolski 2021). Although this is beneficial to the GVCs' participation in a given economy, the internet of things is associated with risk factors such as internet fraud, deliberate viruses attacks, and cyber-attacks linked to the information network that may result in interruption and breakdowns. GVCs suppliers and these can affect both the suppliers and consumers. For instance, DHL used to enlighten its customers concerning positive interruptions to their particular

supply chain (Dobrowolski 2021). Similarly, the innovative technologies in the form of a digital business model have transformed the current business plans built on the traditional assumptions of the GVCs. Thus, the recent properties incorporated in the business models' configuration favor the development of new structures of business solutions not instituted in the traditional value chains (Jabłoński and Jabłoński 2021). For instance, the digital model's platforms link demand and supply and offers economies of scale through network creation with a number of users entering into a relationship in global terms (Jabłoński and Jabłoński 2021). This can create challenges to legal authorities ahead of legal order within the framework of a value system. From both the retailers and consumers' end, the transformation in internet technologies has induced massive improvement of the online business and markets globally, because E-commerce allows consumers to patronize online sales easily in real time systems through a "secured payment system", which is the most important aspect that induced buyers to shop online since there are risks associated with E-commerce (Oláh et al. 2019). Therefore, the trade-off that result in greater benefit must be established at large since each company or country that seeks to achieve sustainable development must strive for a balance with environmental risk for its survival (Bergman 2007; Dabija and Pop 2013).

As many previous studies have investigated, inflation and foreign exchange rates have a clear reciprocal causality (Iqbal 2017; Wang and Lee 2016). The passing through of the exchange rate relates to the mutually sustainable impact that an increase in inflation induces the depreciation of the exchange rate (i.e., more local currency per dollar), which then leads to a rise in inflation. This is especially true in small or emerging open economies where the price of imported products is pushed up by local currency depreciation, thus exacerbating the inflationary process. Therefore, maintaining real depreciation and a minimal price level as well as a lower inflation rate will help to improve the export competitiveness of a nation, which will ultimately improve its growth performance. Appreciation of the lira reduces the costs of importation for industrial inputs as well as manufacturing costs and consequently increases overall supply to a greater level than the decrease in total demand because of the decline in net exports. However, the reduction in the value of the lira induced total demand more than the decline in overall supply.

Nevertheless, to meet the objective of this research, the reviewed literature shows that the findings of this study could be positive, negative, or insignificant with respect to the overall nexus among competitively valued exchange rates, price level, and growth performance in Turkey with new insight from the global value chain.

*Theoretical Framework*

The most extensively used theories on GVCs are organizational theories. Some of these theories have a worldwide scope and are relevant to our current focus on GVCs. The resource-based view and stakeholder theory are the two most commonly used theories in the global value chain. Intriguingly, institutional theory has rarely been applied to international value chain research, despite gaining popularity in sub-supplier research in terms of institutional logistics and entrepreneurial theories (Grimm Jörg and Joerg 2018). Institutional theory, according to Gereffi's basic framework (1994), deals with how chains are defined by their input–output relationships, how they are constructed over space, and how they are managed, including the value exchange rate and pricing. Following in the GVC tradition, subsequent studies have focused on governance, specifically how leading firms coordinate product and information exchange. While the stakeholder theory examines how deep-in-the-supply-chain enterprises and individuals might become main stakeholders of more prominent focal firms. Many of these out-of-town suppliers are, at best, minority stakeholders in either government or companies. It is worth looking into if they can become more relevant as sub-suppliers and multitier connections grow more common (Sarkis et al. 2019).

Another theoretical viewpoint having worldwide implications is that technology can be used to assist economic growth from its environmental consequences, as shown

above in the schematic framework (See Figure 1). However, this decoupling perspective has come under examination and has been criticized in general (Troumbis et al. 2018). Technology, from this theoretical perspective, may have a role beyond decoupling. It may also reveal how technology can help governments and trade merchandise in the global value chain better integrate into the larger global supply chain, particularly with respect to the Turkish economy.

Another theory or model that supports our investigation is the Mundell–Fleming model, which incorporates economic variables such as real GDP, real net exports, real taxes, consumption, and revaluation effects associated with exchange rate changes, among other variables. The model describes the impossible trinity of maintaining a fixed exchange rate, free capital mobility, and independent monetary policy. Therefore, two of the three can be achieved simultaneously, particularly in a small open economy because, according to the Mundell–Fleming model, the economy is not big enough to influence foreign incomes and the global level of interest rate.

However, one of the most striking implications of the model is that uncertainty can affect the first moments of endogenous variables such as the terms of trade, prices, exchange rate, and consumption. For example, a rise in domestic monetary variability raises the prospects that domestic workers will be called on to supply unexpectedly high levels of labor when consumption and prices are high and the desire for leisure greatest. This effect tends to raise relative domestic wages, prices, and lower worldwide consumption. This natural incorporation of uncertainty underprices rigidity suggests that we may finally be close to understanding, at an analytical level, some of the gains monetary unification confers by eliminating exchange rate uncertainty. Those gains are fundamental to the affirmative case for monetary union, as outlined so briefly in Mundell (1961)'s piece.

Despite the shortcomings associated with the Mundell–Fleming model, the model provides some sense with respect to our study on the nexus among competitively valued exchange rates, price levels, and growth performance in the Turkish economy, particularly when directed at participation in the global value chains to the best of the author's knowledge. For this reason, the Mundell–Fleming model also serves as a basis for this study.

### 3. Methodology

In order to draw inference for this study about the time series investigation, a stationarity test is conducted using Augmented Dickey-Fuller (ADF) (Dickey and Wayne 1981), and Phillips-Perron (PP), 1988 to avoid spurious regression and autocorrelation problems, as recommended by Enders (1995). In addition, Ng perron (2001) unit test, which combines four test statistics, namely: (MZa, MZt, MSB, and MPT) is further incorporated due to limitations of ADF and PP tests, which over reject the null hypothesis although it is true or accept it although it is false (Shahbaz and Salahuddin 2009). Accordingly, Bayer and Hanck Cointegration (BHC) is also employed, and the order of the variable has been ascertained based on order one (i.e., I (1)) for the ADF, PP, and Ng perron unit root test in Tables 1 and 2. In the BHC, a cointegration relation is established in the first hypothesis in Table 3. Therefore, an Autoregressive Distributive lag (ARDL) technique along with the Error correction model (ECM) are considered to estimate the impact of the considered variables in the study.

### 3.1. Technique of Data Collection and Specification of the Model

The research employs annual data from 1980–2020 to evaluate the overall nexus among competitively valued exchange rates, price level, and growth performance in Turkey with new insight from the global value chain. Exports, as well as imports, are used as proxies for foreign trade. The sample was chosen to reflect the adjustment strategies based on the export expansion and market forces mechanism adopted in Turkey during the study period. The data have been collected from the database of the World Bank. However, the multivariate unit of measurement includes GDP in constant Local Currency Unit, Real effective exchange rate index of (2010 = 100), and it is included given its role in

inducing macroeconomic activities in Turkey. The exchange rate is expected to influence the economic performance largely when the level of prices is unfavorable, exports (% of GDP) and imports (% of GDP). The model is logged linearized and specified as below:

$$\ln\text{RGDP}_t = \beta_0 + \beta_1 \ln\text{REER}_t + \beta_2 \ln\text{EXPT}_t + \beta_3 \ln\text{IMPT}_t + \varepsilon_t \tag{1}$$

lnRGDP = Economic growth
lnREER = Real effective exchange rate
lnEXPT = Export
lnIMPT = Import

The t subscript indicates time, $\beta_0$ is the constant term, while $\beta_i$ (i = 0, 1 . . . 4) represents the coefficients of the explanatory variables. The signs of $\beta_1$ and $\beta_2$ are both expected to be positive. The expected sign of $\beta_3$ should be negative, as documented in the literature. While $\varepsilon_t$ is the white noise that takes into account the impacts of series not included in our model and should satisfy the assumption of normally and independently distributed around zero mean and constant variance (i.e., $\varepsilon_t \sim$ NIID (0, 1)).

### 3.2. Bounds Test for Cointegration

This study will apply the ARDL bounds test cointegration technique developed by Pesaran et al. (2001) as the best and most realistic model using the F-statistic to establish the significance of the coefficients of the considered lagged variables.

$$\Delta ln\text{RGDP}_t = \beta_0 + \sum_{i=1}^{p} \beta_{1i}\Delta ln\text{RGDP}_{t-1} + \sum_{i=0}^{p} \beta_{2i}\Delta ln\text{REER}_{t-1} + \sum_{i=0}^{p} \beta_{3i}\Delta ln\text{EXPT}_{t-1} + \sum_{i=0}^{p} \beta_{4i}\Delta ln\text{IMPT}_{t-1} + \lambda_1 ln\text{RGDP}_{t-2}$$
$$+\lambda_2 ln\text{REER}_{t-2}\lambda_3 ln\text{EXPT}_{t-2} + \lambda_4 ln\text{IMPT}_{t-2} + \varepsilon_{it} \tag{2}$$

where $\varepsilon_t$ denotes a white noise and error term, first difference operator ($\Delta$) denotes log of the variable. However, $H_0$: $\lambda_i = 0$, while $H_i$: $\lambda_i \neq 0$, i = 1, 2, 5 of no cointegration among the variables. It is important to note that sets of critical values for comparison have been reported by Pesaran et al. (2001). Overall, when the calculated "F-statistics" value is greater than both lower (i.e., I(0)) and upper bounds (i.e., I(1)), then there is cointegration among the variables, and $H_0$: = 0 of no cointegration is rejected. When the f-values are between the upper and lower bounds, the decision is indecisive. If the considered variables are found to be cointegrated, we can run the ECM equation shown below:

$$\Delta ln\text{RGDP} = \beta_0 + \sum_{i=1}^{p} \beta_{1i}\Delta ln\text{RGDP}_{t-1} + \sum_{i=1}^{q} \beta_{1i}\Delta ln\text{REER}_{t-1} + \sum_{i=1}^{r} \beta_{1i}\Delta ln\text{EXPT}_{t-1} + \sum_{i=1}^{s} \beta_{1i\Delta} ln\text{IMPT}_{t-1} + \pi_1 ECT_{t-1} + \varepsilon_{1t}, \tag{3}$$

$$\Delta ln\text{REER} = \beta_0 + \sum_{i=1}^{q} \beta_{1i}\Delta ln\text{REER}_{t-1} + \sum_{i=1}^{p} \beta_{1i}\Delta ln\text{RGDP}_{t-1} + \sum_{i=1}^{r} \beta_{1i}\Delta ln\text{EXPT}_{t-1} + \sum_{i=1}^{s} \beta_{1i\Delta} ln\text{IMPT}_{t-1} + \pi_2 ECT_{t-1} + \varepsilon_{2t}, \tag{4}$$

$$\Delta ln\text{EXPT} = \beta_0 + \sum_{i=1}^{r} \beta_{1i}\Delta ln\text{EXPT}_{t-1} + \sum_{i=1}^{q} \beta_{1i}\Delta ln\text{REER}_{t-1} \sum_{i=1}^{s} \beta_{1i\Delta} ln\text{IMPT}_{t-1} + \sum_{i=1}^{p} \beta_{1i}\Delta ln\text{RGDP}_{t-1} + \pi_3 ECT_{t-1} + \varepsilon_{3t}, \tag{5}$$

$$\Delta ln\text{IMPT} = \beta_0 + \sum_{i=1}^{s} \beta_{1i}\Delta ln\text{IMPT}_{t-1} + \sum_{i=1}^{q} \beta_{1i}\Delta ln\text{REER}_{t-1} \sum_{i=1}^{r} \beta_{1i\Delta} ln\text{EXPT}_{t-1} + \sum_{i=1}^{p} \beta_{1i}\Delta ln\text{RGDP}_{t-1} + \pi_4 ECT_{t-1} + \varepsilon_{4t}, \tag{6}$$

$\varepsilon_{1t}$, $\varepsilon_{2t}$, $\varepsilon_{3t}$, and $\varepsilon_{4t,}$ are white noise ($\varepsilon_t \sim$ NIID (0, 1)).
The $ECT_{t-1}$, is the speed of adjustment to the restoration path. The sign of ECT should be negative and statistically significant between (0, 1).

### 3.3. Stability Test

Determining the best fit of the model requires that various stability and diagnostic tests be performed. This includes the Ramsey RESET test. One of the ways of detecting structural break is by plotting the recursive Cusum and Cusum of squares tests cited in (Brown et al. 1975) following Adedoyin et al. (2021). The decision rules that if the plots

deviate from the 5% significance bounds, the presence of structural break(s) exists in the model. On a final note, ARDL has information concerning the structural break in time series data (Shahbaz and Salahuddin 2009). Yet, "appropriate modification of the orders of the ARDL model is sufficient to simultaneously correct for residual serial correlation and the problem of endogenous variables" (Pesaran et al. 2001), cited in (Shahbaz and Salahuddin 2009). Therefore, the model in this study is within the confidence bounds. Further tests include serial correlation, heteroscedasticity, and Jarque-Bera normality.

## 4. Results

*Stationarity Tests*

It is recommended that the stationarity of the variables of interest be tested in empirical studies. However, estimating the ARDL requires that none of the variables are I(2). Enders (1995) recommended that researchers employ both the (ADF) test (1981) and (PP) test (1988) of stationarity. However, the PP test determines the robustness of the ADF test.

As shown in Table 1, the $MZ_a$ and $MZ_t$ tests hypothesis is established as unit root, while the MSB and MPT tests are established as stationary. The result from Table 1 revealed that the test statistics of $MZ_a$ and $MZ_t$ tests are smaller than the critical values calculated by (Ng and Perron 2001) for all the variables. Correspondingly, Table 1 revealed that the test statistics of MSB and MPT tests are greater than the critical values calculated by (Ng and Perron 2001) for all the variables. The real GDP, real effective exchange rates, exports, and imports are integrated into the first order. As the results are consistent with the I(1) cointegrating order for all the series, this indicates that the degree of integrating problem does not exist among the study variables because the results are reversed.

**Table 1.** Ng Perron Unit root at Level.

| | **At Level** | | | | **At First Difference** | | | |
|---|---|---|---|---|---|---|---|---|
| **Variables** | $MZ_a$ | $MZ_t$ | **MSB** | **MPT** | $MZ_a$ | $MZ_t$ | **MSB** | **MPT** |
| lnRGDP | 1.83783 | 2.94444 | 1.60213 | 197.035 | −19.3894 | −3.09663 | 0.15971 | 1.32415 |
| lnEXPT | −0.30930 | −0.19469 | 0.62945 | 24.5696 | 1.18841 | 3.26535 | 2.74767 | 499.228 |
| lnREER | −1.91433 | −0.77035 | 0.40241 | 10.5300 | −6.43326 | −1.79320 | 0.27874 | 3.80933 |
| lnIMPT | −0.75727 | −0.36387 | 0.48050 | 15.7542 | 19.3452 | −3.10938 | −0.16073 | 1.26892 |

As shown in Table 2, it can be seen that the outcome of both the ADF and PP for lnRGDP, lnREER, lnEXMPT, and lnIMPT is of order I(1). This means that the variables are first difference stationary. For that reason, the test satisfies the condition for the application of the ARDL bounds test for the cointegration technique since none of the variables are I(2). This result is in line with Faisal et al. (2017).

**Table 2.** ADF and PP Stationarity Tests.

| | **ADF** | | **PP** | |
|---|---|---|---|---|
| **Variables** | **Test-Statistics Level** | **Test-Statistics First Diff.** | **Test-Statistics Level** | **Test-Statistics First Diff.** |
| lnRGDP | −0.389949 | −6.514453 | | −6.635688 |
| lnEXPT | −4.697329 | −6.257480 | −4.711909 | −6.515829 |
| lnIMPT | −2.473180 | −5.859016 | −2.609744 | −7.097405 |
| lnREER | −4.062896 | −5.031149 | −3.183120 | −4.273850 |

As shown in Table 3, the value of the F-statistic (12.12496 ***) is higher than the upper and lower bounds values. The results indicate that there is a long-run association between the study variables of interest, as supported by Fidan (2006); Ergin and Filiz (2017); Bahmani-Oskooee et al. (2017); Faisal et al. (2017); Ozdeser et al. (2021). The optimal lag has been automatically selected by the AIC criteria. The critical values have been established based on Pesaran et al. (2001).

**Table 3.** Bounds Test for Cointegration Results.

| Model | ARDL (4, 4, 3, 4) | | | |
|---|---|---|---|---|
| Optimal Lag | 3 | | | |
| F-Statistics (Bounds test) | 12.12496 *** | | | |
| Critical—values | At 1% | At 2.5% | At 5% | At 10% |
| Lower bound I (0) | 3.65 | 3.15 | 2.79 | 2.37 |
| Upper bound I (1) | 4.66 | 4.08 | 3.67 | 3.2 |

*** indicates significance at all levels; 1%, 2.5%, 5% and 10%.

As shown in Table 4, the BH cointegration permits us to combine different cointegration test results to draw a robust and conclusive finding. Therefore, the EGT-JOT and EG-JOT-BOT-BAT Fisher statistics are greater than the critical values at a 5% significance level when we utilize real effective exchange rate, exports, and imports as dependent variables in this model. The hypothesis of no cointegration among the variables is rejected. It can be concluded that a long-run relationship exists among real effective exchange rate, exports, and imports, which can induce GVC competitiveness and real economic performance in the long run in the case of Turkey. This result is in line with Mohammed et al. (2020).

**Table 4.** Result of Bayer and Hanck (2013) Test.

| **Fisher F-Statistics** | | | **Cointegration** |
|---|---|---|---|
| | EGT-JOT | EGT-JOT-BOT-BAT | |
| | 14.894 | 20.486 ** | Cointegrated |
| Statistical significance at 5% | 10.630 | 17.682 | |

** indicate significance at 5% level respectively.

As shown in Table 5, the short-run coefficients show that lnRGDP, lnREER, lnEXPT, and lnIMPT are statistically significant and are induced by their lags. For instance, the relationship between D(lnRGDP) in the short run in both the first and the second lag has an endogenously positive impact on the dependent variable, but it is self-adjusting at 10% with the third lag. Contrastingly, D(lnREER) has a positive and significant impact on the dependent variable at 1% up to two lags. By implication, this means that appreciation in the exchange rate (i.e., Turkish lira) has a positive economic impact on the growth performance of the Turkish economy and it is induced by the lags effect. This finding is in line with the result of (Bahmani-Oskooee et al. 2017; Ergin and Filiz 2017; Fidan 2006). Accordingly, the D(lnEXPT) also reveals a correct sign with a positive and statistically significant impact on the performance of the Turkish economy. The implication for this is that increase in the export base of the economy can induce positive and significant benefits to the economic growth dynamics of the Turkish economy by two lags. On the other hand, D(lnIMPT) can positively affect economic performance when there is an increase in the world-class imported inputs for industrial consumption. However, the D(lnIMPT) is increasingly related to the economic performance, and it also affects the growth performance by up to two lags. The implication for this is that, increase in the normal import of goods and services has a negative effect on the economic performance of the Turkish economy as supported by the literature and the economic theory based on a priori criteria (see, Koutsoyiannis 1977).

**Table 5.** Short-Run Relationship along with ECM.

| Variable | Estimate | Standard Error | T-Value | Probability |
|---|---|---|---|---|
| D(lnRGDP (−1)) | 0.084156 | 0.120309 | 0.699495 | 0.4932 |
| D(lnRGDP (−2)) | 0.123677 | 0.109842 | 1.125949 | 0.2750 |
| D(lnRGDP (−3)) | −0.221043 | 0.109829 | −2.012605 | 0.0594 * |
| D(lnREER) | 0.070232 | 0.020201 | −3.476611 | 0.0027 *** |
| D(lnREER (−1)) | 0.016175 | 0.019519 | 0.828714 | 0.4181 |
| D(lnREER(−2)) | −0.061059 | 0.018480 | −3.304057 | 0.0039 *** |
| D(lnREER(−3)) | −0.025803 | 0.018480 | −1.430308 | 0.1698 |
| D(lnEXPT) | 0.207121 | 0.060887 | 3.401727 | 0.0032 *** |
| D(lnEXPT (−1)) | −0.000683 | 0.061287 | −0.011139 | 0.9912 |
| D(lnEXPT (−2)) | 0.201274 | 0.044887 | 4.483991 | 0.0003 *** |
| D(lnEXPT (−3)) | −0.184381 | 0.036516 | −5.049385 | 0.0001 *** |
| D(lnIMPT) | 0.021102 | 0.044406 | 0.475205 | 0.6404 |
| D(lnIMPT (−1)) | −0.333924 | 0.062851 | −5.312969 | 0.0000 *** |
| D(lnIMPT (−2)) | −0.361814 | 0.062946 | −5.748056 | 0.0000 *** |
| C | $785\,8 \times 10^3$ | $5.09\,8 \times 10^3$ | 1.541101 | 0.1441 |
| ECT (−1) | −0.102415 | 0.011898 | −8.607959 | 0.0000 *** |

*, *** indicate significance at 10% level and 1% level respectively.

However, the error correction term in Table 5 revealed a correct sign with a value—0.1024 (i.e., approximately) and a significant corresponding probability of 0.0000 (i.e., 1%). Thus, the −0.1024 implies a 10% speed of adjustment to the equilibrium in the subsequent period. By implication, the system is capable of converging towards an equilibrium path after some shocks to the system. Therefore, the competitiveness of the Turkish economy's exports in the global value chain (GVCs) requires at least a 10% share of value-added in exports annually. This has similarities to the results obtained by (Ergin and Filiz 2017), (Bahmani-Oskooee et al. 2017; Faisal et al. 2017; Mohammed et al. 2020; Ozdeser et al. 2021; Fidan 2006). As a result, the restoration to the equilibrium path will take at least ten years unless equilibrium agents (i.e., government and monetary authorities) implement active policies that will accelerate the export competitiveness. The restoration to the equilibrium in the subsequent period has a minimal effect on the system. However, considering the recent lira depreciation and developing nature of the Turkish economy, our model results make some sense, as supported by the 2016 World Bank report on Turkey, which stressed that growth in Turkey would decline from 6.1% in 2015 to 2.1% in 2016.

As shown in Table 6, the real effective exchange rate negatively affects economic performance in real terms. However, the export of goods positively affects economic performance in real terms, but is insignificant to induce any effect on the global value chain. In a GVC world, the competitiveness of an economy's exports emanates from imported inputs embedded in their own previous exports, and an increase in imported inputs (i.e., in GVC) has a positive effect on economic performance in real terms in the long run in the Turkish economy since effective exports depend on world-class imported inputs, as supported by Ergin and Filiz (2017); Faisal et al. (2017); Mohammed et al. (2020); Ozdeser et al. (2021).

**Table 6.** The Long-Run Coefficients.

| Variables | Estimate | Standard Error | T-Value | Probability |
|---|---|---|---|---|
| lnREER | −0.271676 | 0.101964 | −2.664425 | 0.0158 |
| lnEXPT | 1.712415 | 1.022736 | 1.674347 | 0.1113 |
| lnIMPT | 2.685797 | 0.987151 | 2.720757 | 0.0140 |
| C | 13.911154 | 2.963600 | 4.694006 | 0.0002 |

As shown in Table 7, the result of the LM-Test shows that the probability of Chi-square is larger than 5%. For that reason, the null hypothesis cannot be rejected. By implication, this means that the residuals are not serially correlated, and this is good for the model in

this study. However, the Breusch-Godfrey Heteroskedasticity test result shows that the probability value is 0.1740, and its corresponding F-value is 1.569035, which is higher than the expected *p*-value of 0.05. It can therefore be concluded that there is no evidence of Heteroskedasticity. The Ramsey Reset results also show that the probability value is greater than 0.05%. This indicates that the model is fit, and the explanatory variables can explain the response variable. Similarly, as shown in Table 7, the distribution of the model is normal because the residuals are normally distributed around zero mean and constant variance based on the corresponding probability value of Jarque Bera 1.069 (0.586 approximately), as supported by Ozdeser et al. (2021); Faisal et al. (2017).

**Table 7.** Residual Diagnostic Checks.

| Statistical Tests/J-B Stat. | | Probability | |
|---|---|---|---|
| B-G Serial Corr. Lm-Test F-statistics | 0.295424 | Prob. F (2,26) | 0.7482 |
| Heteroskedasticity Test (Harvey) | 1.317675 | Prob. Chi-square (2) | 0.5175 |
| F-statistics | 1.569035 | Prob. F (5,28) | 0.1740 |
| Obs R-square | 22.59770 | Prob. Chi-Square (5) | 0.2065 |
| Normality Test | 1.069070 | - | 0.585942 |
| Scaled explained SS | 32.89948 | Prob. Chi-Square (5) | 0.0172 |
| Ramsey Reset Test F-statistics | 0.070225 | Prob. | 0.7942 |

As shown in Figure 2, the stability plots provide valuable insight into the stability of the model. The plots fall within the accepted area, which shows that the parameters are stable and the variance estimates are within the 5% significance boundary. This indicates the stability of our model. This result is supported by Faisal et al. (2017); Ozdeser et al. (2021). Furthermore, one of the ways of detecting structural break is by plotting the recursive Cusum and Cusum of squares tests, as with Adedoyin et al. (2021). The decision rule is that if the plots deviate from the 5% significance bounds, the presence of structural break exists in the model. ARDL model has information concerning the structural break in time series data (Shahbaz and Salahuddin 2009). Yet, "appropriate modification of the orders of the ARDL model is sufficient to simultaneously correct for residual serial correlation and the problem of endogenous variables" (Pesaran et al. 2001) cited in (Shahbaz and Salahuddin 2009). Therefore, the model in this study is within the confidence bounds.

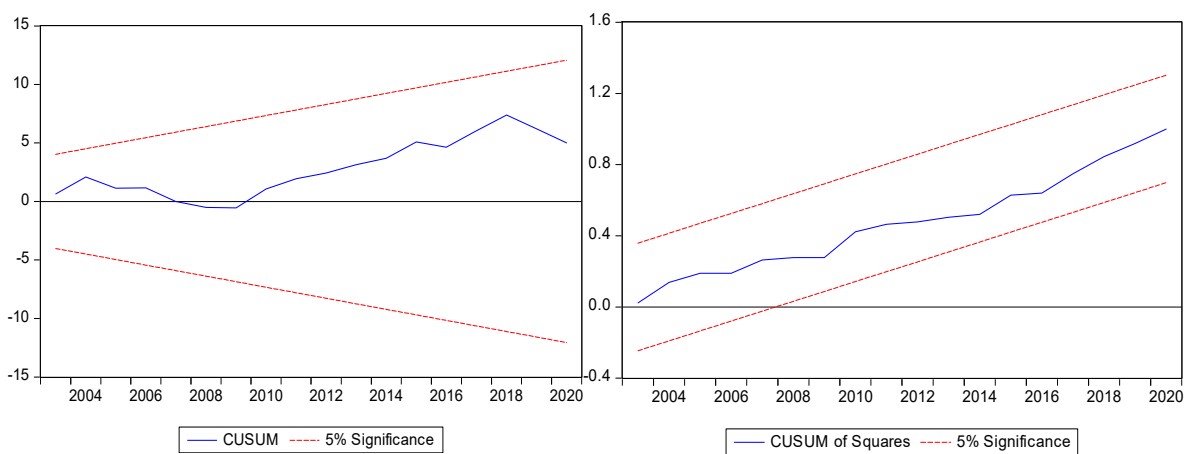

**Figure 2.** Stability Test of Cusum and Cusum of squared plots.

## 5. Discussion

Nowadays, global economies are increasingly participating in the global value chains (GVCs). The GVCs continue to play a significant role in the global macroeconomic dynam-

ics of these countries the world over. From the managerial and government perspectives, GVCs offer even further hopes and opportunities both from the standpoint of achieving a growing share of external trade and the GDP and employment welfare targets of the various governments in global economies. For example, the emergence of GVCs has significantly shaped world trade particularly in areas as diverse as commodities, electronics, and business services outsourcing, among other areas since the firms and economies involved in the chains hold some value and benefits from the exports of the finished products. By implication, GVCs are a significant means for tracing the evolving trends of global production in the current globalized economy with very complex industry interactions, connecting geographically distributed activities and actors of a single industry, and assessing the roles they play in both developed and developing economies. Thus, the structure of the GVCs focuses on the value-added systems within a sector, from conception to output and final use. It discusses job descriptions, requirements, legislation, products, procedures, and markets, in particular industries and locations, offering a holistic view of both "top-down" and "bottom-up" global industries (Gereffi 1994) and (Globalvaluechains.org 2011).

Next, we scrutinize the empirical result. The discussion starts from Tables 1 and 2, because they gave the study a road map to proceed with the subsequent analyses. For instance, the result from Table 1 showed that the NG-Perron unit root test statistics of $MZ_a$ and $MZ_t$ tests are smaller than the critical values calculated by (Ng and Perron 2001) for all the variables. Correspondingly, Table 1 revealed that the test statistics of MSB and MPT tests are greater than the critical values calculated by (Ng and Perron 2001) for all the variables. By implication, the real GDP, real effective exchange rates, exports, and imports are integrated of the first order and are in line with the I(1) cointegrating order for all the series. Therefore, the extent of the integrating problem does not exist among the study variables because the results are reversed. The results from Table 2 also revealed the outcome of both the ADF and PP unit root tests for lnRGDP, lnREER, lnEXMPT, and lnIMPT based on order I(1). This means that the variables are first difference stationary. For that reason, the test satisfies the condition for the application of the ARDL bounds test for cointegration technique since none of the variables are I(2), as supported by the findings of Faisal et al. (2017).

Next, we scrutinize the ARDL Bounds approach to cointegration findings in Table 3. The findings reveal that the computed value of F- statistics of 12.12 is higher than the upper bound table value, which is 4.66. Thus, the null hypothesis of no cointegration among the variable of interest in the study is rejected. The implication for this as shown by the F- statistics, as lnRGDP, lnEXPT, lnREER, and lnIMPT are bound together because of equilibrium forces towards a long-run relationship. In addition, these variables offer information relevant to the external trade competitiveness along with GVCs for the Turkish economy in the long run. For example, previous literature has shown that real GDP, exports, exchange rate, and imports may not always be linked, but they are related. Thus, the exchange rate has a larger effect on exports and imports in the long run than in the short run. Also, changes in terms of trade and exchange rate have positive effects on the industrial production index of the Turkish economy (Fidan 2006; Bahmani-Oskooee et al. 2017); Faisal et al. (2017); Ozdeser et al. (2021). By implication, fluctuations in the exchange rate and terms of trade positively affect the economic growth of the Turkish economy (Ergin and Filiz 2017).

Next, we examine the result of the Bayer and Hanck combined cointegration in Table 4, which gives us different cointegration test results to draw a robust and conclusive finding. Therefore, the EGT-JOT and EG-JOT-BOT-BAT Fisher statistics is greater than the critical values at 5% significance level when we utilize real effective exchange rate, exports and imports as dependent variables in this model. The hypothesis of no cointegration among the variables is rejected. The implication for this is to validate the bound test approach to the cointegration test and the result is trustable with the bound test result. Therefore, a long-run relationship exists among real effective exchange rate, exports, and imports,

which can induce GVC competitiveness and in real economic performance in the long run in the case of Turkey. This result is in line with Mohammed et al. (2020).

Next, we examine Tables 5 and 6, the estimates of lnEXPT, lnREER, and lnIMPT on lnRGDP, in both the short and long run. The results revealed that the overall estimates from the short run and the long run are statistically significant, and the lnEXPT, lnREER, and lnIMPT are induced by their lags. This means that the exchange rate affects the economic performance in Turkey with lag and this is supported by (Bahmani-Oskooee et al. 2017). However, the error correction term is revealed to have the correct sign with a value −0.1024 (i.e., approximately), and its corresponding probability is 0.0000, which is statistically significant at 1%. The −0.1024 implies a 10% speed of adjustment to the equilibrium in the subsequent period. By implication, the system is capable of converging towards an equilibrium path after some shocks to the system. Therefore, the competitiveness of the Turkish economy's exports in the global value chain (GVC) requires a share of at least 10% of value-added in exports annually. This is in line with the results obtained by (Ergin and Filiz 2017; Bahmani-Oskooee et al. 2017; Faisal et al. 2017; Mohammed et al. 2020; Ozdeser et al. 2021; Fidan 2006). As a result, the restoration to the equilibrium path will take at least ten years, unless equilibrium agents (government and monetary authorities) implement active policies that will accelerate the export competitiveness, because the restoration to the equilibrium in the subsequent period has a minimal effect on the system. However, considering the recent lira depreciation and developing nature of the Turkish economy, our model results make some sense, as supported by the 2016 World Bank report on Turkey, which stressed that growth in Turkey would decline from 6.1% in 2015 to 2.1% in 2016. Next, we examine the results of the long run. From Table 6, the real effective exchange rate negatively affects the economic performance in real terms. However, the export of goods positively affects economic performance in real terms, but is insignificant to induce any effect on the global value chain. In a GVC world, the competitiveness of an economy's exports emanates from imported inputs embedded in their own previous exports, and an increase in imported inputs (i.e., in GVC) has a positive effect on economic performance in real terms in the long run in the Turkish economy since effective exports depend on world-class imported inputs, as supported by Ergin and Filiz (2017); Faisal et al. (2017); Mohammed et al. (2020); Ozdeser et al. (2021).

Following are the robustness checks and diagnostic tests that fit the model and satisfy the objective of this study. For example, the residuals of the model are homoskedastic and serially uncorrelated. Furthermore, the errors of the model are normally independently and identically distributed around zero mean and constant variance (i.e., NIID-(0, 1)) and the functional form of the model is correctly specified, as supported by Ozdeser et al. (2021); Faisal et al. (2017). Finally, the coefficients of the model are also stable. The study scrutinized the structural breaks by means of the cumulative sum of squares to ascertain the stability of the parameters in the estimated model, which is exhibited in Figure 2, which follows (Adedoyin et al. 2021; Faisal et al. 2017; Ozdeser et al. 2021). The test statistic for the estimated parameters should be within the significant bounds of the 95% confidence interval, and this was confirmed. Over time, Figure 2, also confirmed the stability of the estimated coefficients as recommended, see (Brown et al. 1975; Shahbaz and Salahuddin 2009).

## 6. Conclusions

The emergence of GVCs has impacted the global trade in diverse areas for both the imports and exports of products such as electronics, commodities, and business service outsourcing, among other areas, since the countries involved in the GVCs retain some value(s) and benefit(s) from the exports of the finished product traded in the GVCs world. Nowadays, global value chains (GVCs) are increasingly shaping the global economy, covering a growing share of international trade, GDP, and employment globally.

In this study, an ARDL model along with ECM and the newly Bayer and Hanck (2013) combined cointegration test has been applied to investigate the nexus among competitively valued exchange rates, price level, and growth performance in the Turkish economy; new

insight from the GVCs is investigated using annual data from 1980 to 2020. The following conclusions are drawn from the research findings: Firstly, a cointegrating relationship exists in both the short run and the long run among the variables of interest. Namely, real effective exchange rate, exports, and imports. This cointegrating relation is confirmed by the ARDL bounds test approach with a value of F- statistics (i.e., 12.12496 ***) greater than the critical values at all significance levels. Secondly, while real effective exchange rate, exports, and imports promote economic performance and external trade competitiveness mainly when directed at GVCs in both the short and long run, an increase in exchange rate fluctuations and high imports prices can negate the economic performance of the Turkish economy. The study finds that the Turkish economy is favorable mainly for industrial and commercial investments and projects because exports reveal a positive effect on economic performance. Still, it is insignificant to induce any effect on the global value chain. The coefficient of $Ect_{-1}$ is negative (i.e., $-0.1024$ approximately) and statistically significant (i.e., 0.0000). Since our results are robust and the finding from ECM shows that the model's lagged coefficient of the ECT is negative and statistically significant at 1%. Thus, the speed of adjustment requires 10% annually for the convergence to the long-run equilibrium to be met. Overall, the study concludes that there is a long-run relationship among competitive valued exchange rate, price levels, and economic performance in the Turkish economy, particularly when directed at achieving competitiveness in GVCs. Although our findings are unique in terms of the robustness of our results and the GVCs, they are similar to the findings of Ergin and Filiz (2017), (Kandil 2004; Kandil and Mirzaie 2005; Tadesse 2009; Bahmani-Oskooee and Kandil 2007; Nabli and Véganzonès-Varoudakis 2004). Similarly, Joof and Jallow (2020) find a long relationship between the exchange rate, price levels, and economic performance.

### 6.1. Policy Implications

The policy implication of our findings is the fact that our results are robust because they reject the null hypothesis of the no long-run relationship and accept the alternative hypothesis of the significant impact of exports, real effective exchange rate, and imports on economic performance and external trade competitiveness, particularly when directed at GVCs in both the short and long run in the Turkish economy. However, the managerial implications for the long-run association among the exports, real effective exchange rate, and imports have been mixed. The fact is that GVCs revealed significant managerial prospects and opportunities that can transform Turkish industrial sectors, particularly the Agric sector, into a competitive sector, particularly when the value-added production is integrated into current GVCs world markets because each firm or country retains some certain value and benefits related to the consumption of the final product.

Thus, development in economic performance, which is measured by the lnRGDP, can enhance economic activities and an export-based of economy. Therefore, exchange rate volatility can be minimized in the economy through external trade policy directed at industrial sectors export capacity and Agric sector operations by farmers who grow crops and processors among others. Furthermore, for the managerial contributions of the study in terms of the generalization of the findings are unique and this study argues that global dynamics, GVCs dynamics, and changing industry accompanied with consumer dynamics offer prospects for innovation to the industrial sector (i.e., managerial settings), the entire agriculture ecosystem to face a significant change that is revolutionizing the future of exports from the farmers who grow crops, from producers to suppliers and processors to retailers and consumers. This is hardly covered by the previous studies in the literature. For instance, the harvest system can be essentially transformed from a commodity-based export meant for industrial inputs to an improved and global value-added supply chains consumption. Thus, farmers and producers are challenged with a higher participation burden across the GVCs world. The implication for these is that firms can claim some value and benefits in the final products, although they may need guarantees on the value creation, transparent operations, corporate social responsibility,



and profit enhancement that induce positive transformation of the firms and agriculture operations before taking risk efforts that help in achieving the future gains. Theoretically, our study is guided by the Mundell–Fleming model and stakeholder theory. Therefore, the Mundell–Fleming model incorporates economic variables such as real GDP, real net exports, real taxes, consumptions, and revaluation effects associated with exchange rate changes, among other variables. The model describes the impossible trinity of maintaining a fixed exchange rate, free capital mobility, and independent monetary policy. Therefore, two of the three can be achieved simultaneously, particularly in a small open economy because, according to the Mundell–Fleming model, the economy is not big enough to influence foreign incomes and the global level of interest rate. However, one of the most striking implications of the model is that uncertainty can affect the first moments of endogenous variables such as the terms of trade, prices, exchange rate, and consumption. For example, a rise in domestic monetary variability raises the prospects that domestic workers will be called on to supply unexpectedly high levels of labor when consumption and prices are high and the desire for leisure greatest. This effect tends to raise relative domestic wages, prices, and lower worldwide consumption. This natural incorporation of uncertainty underprices rigidity suggests that we may finally be close to understanding, at an analytical level, some of the gains monetary unification confers by eliminating exchange rate uncertainty. The stakeholder theory examines how deep-in-the-supply-chain enterprises and individuals might become the main stakeholders of bigger focal firms. This contribution is rarely considered by the previous studies, and this study filled this gap. Theoretically, depreciation of the domestic currency of the Turkish economy due to fluctuations in exchange rates can induce the external trade competitiveness of domestic exports foreigners to find our exports cheaper while our industrial sectors find their exports dearer, which can lead to higher prices in the Turkish economy.

Based on the objective of this study, the study recommends that policies enhancing a 10% equilibrium convergence are required annually to competitively minimize the dependence on foreign value-added inputs by importing only world-class inputs for value addition and exports benefits in the competitive GVCs world. In addition, policymakers and relevant stakeholders should design agricultural value addition production strategies and an inclusive policy framework that includes various sectors and action areas that can offer rooms for opportunities and discussions and enhance the Agric and industrial sectors to retain some value and benefits in the final products in the GVCs market.

### 6.2. Limitations

Against this background, this study is limited in terms of qualitative data that could be used primarily by directly engaging with the firms and industrial sectors in the Turkish economy to offer policy inferences. As a result, we employed available secondary data for this empirical study. However, an extensive inclusion of data such as cross-sectional or panel would have reinforced further justifications of the impact of exports, real effective exchange rate, and imports on economic performance and external trade competitiveness mainly when directed at GVCs in both the short and long run in the Turkish economy. Even though the study variables fit the study's model and the inclusion of white noise (i.e., $\varepsilon_t$) captures the effects of other variables not included in the model, it is possible that other variables from the literature such as distribution of value-added in the value chain, foreign value-added shares, domestic value-added, bilateral value-added exports, and value-added for final consumption among other indicators would have further justified the research findings and conclusions.

However, future studies should fill this gap with micro qualitative data. Furthermore, future research should focus on agriculture value-addition due to the role it plays in the GVCs process. Furthermore, future studies should also include concentration on studying agriculture value-addition into the competitive GVCs world. In addition, future studies should concentrate on investigating monetary policy, GVCs, and economic growth nexus.

**Author Contributions:** Conceptualization, U.A.S., M.A.M.U. and B.Ç.; methodology, U.A.S. and M.A.M.U.; software, U.A.S. and M.A.M.U.; validation, U.A.S., M.A.M.U. and B.Ç.; formal analysis, U.A.S. and M.A.M.U.; investigation, U.A.S. and M.A.M.U.; data curation, U.A.S. and M.A.M.U.; writing—original draft preparation, U.A.S.; writing—review and editing, M.A.M.U.; visualiza-tion, U.A.S. and M.A.M.U.; supervision, B.Ç.; project administration, U.A.S., M.A.M.U. and B.Ç.; funding acquisition, U.A.S., M.A.M.U. and B.Ç. All authors have read and agreed to the published version of the manuscript.

**Funding:** This research received no external funding.

**Institutional Review Board Statement:** Not applicable.

**Informed Consent Statement:** Not applicable.

**Data Availability Statement:** Data and materials used in this article are available from the authors upon request.

**Acknowledgments:** We would like to commend the editors and anonymous reviewers. For providing insightful and constructive comments on an earlier version of this paper.

**Conflicts of Interest:** The authors declare no conflict of interest regarding the publication of this paper.

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
