# Peer review of "The Nexus among Competitively Valued Exchange Rates, Price Level, and Growth Performance in the Turkish Economy; New Insight from the Global Value Chains"

_jrfm, doi:10.3390/jrfm14110528_

Round 1

Reviewer 1 Report

The article requires proofreading. For example, the noun phrase Turkish economy seems to be missing a determiner before it. Consider adding an article.

Although interesting, the introduction to the article is a kind of description of Turkey's economic success. However, there is no clear indication of the research gap or formulation of research problems. The reader has only the opportunity to get acquainted with the purpose of the research. The authors have written, "Our significant contribution to the literature and research analysis/plan is to establish ...." but it seems that it is not enough. The research gap resulting from the analysis of the research conducted so far should be indicated more precisely.

Moreover, the authors' statement about the control of corruption in Turkey is quite surprising. The research results, including data from Transparency International, indicate a severe threat of corruption in Turkey. In this respect, the article also requires thorough rethinking and analysis.

Comments on the second point. The literature review is primarily local (authors mainly from East Asia Region). There is nothing wrong with that. However, whether no one else in the world has researched in the scope presented by the authors? It is not the case of Turkey, but the relationship between the variables indicated by the authors. Please ensure a deep analysis of the literature.

If the research data comes from a normal distribution, the JB statistics asymptotically have a chi-squared distribution.  For a small sample size, the chi-squared approximation is overly sensitive, often rejecting the null hypothesis when it is true. Please provide more information on whether any constraints of JB statistics occurred and implemented preventive measures. 

There are no results of studies confirming that corruption is under control in Turkey. This variable is crucial, especially in emerging markets. There is a need for a very significant expansion of research on this variable.

Conclusion. Please precise indicate the contribution of your research to the development of the discipline.  What previous research results have been confirmed or not by the authors? What are the research limitations? What should be the directions of further research?

Author Response

Reply to Reviwer-1:

We would like to take this opportunity to thank the reviewer-1 of our manuscript for the crucial ideas and points which she/he has raised, below are the authors’ comments of each and every point raised by the reviewer-1.

  • The authors have considered the reviewer comment and corrected the introduction part by (i) further explaining the relationship between the variables under study as well as highlighting the research gap. (ii) Authors have explained the contribution of the study to the literature. (iii) Variable selection and theoretical modelling have been explained. (iv) the aim of the study has been corrected to align it with the topic and the model used. See the points highlighted yellow in the introduction part.
  • Reviver comment on the statement about the control of corruption in Turkey was mis specified. We have made necessary corrections to make it more understandable and clearer. According to the reviewers’ comments more local literature added to ensure the deep analysis of the case of Turkey.
  • The authors have considered the reviewers comment on the precise of the research. The research results, limitations and directions for further research have been added to the conclusion part.
  • The manuscript was submitted for professional English Proof reading as per the reviewer’s comment.

Please see attachment as an revised version of the manuscript. 

Reviewer 2 Report

The topic is interesting and the paper follows the standard research structure.

A few points to consider:

  • a reduction of the abstract. The abstract is supposed to give a brief presentation of the paper rather than a thorough explanation. The details can be found inside the paper and in the conclusions section.
  • a better definition of the research question. The introduction part looks like a long presentation of the macroeconomic developments in Turkey and seems less connected with the objective of the paper. The reader needs to be introduced faster in the research topic, i.e. the relevance of this study should be presented a bit clearer in the introduction section
  • The first part of the Literature Review is a huge paragraph. The ideas presented there are interesting but they need further refinement, and a better connection with the research topic
  • A reference in text to the tables with results for the stationarity and cointegration tests would be important. In this respect section 3.2 maybe needs to be eliminated (looks redundant with section 4.1)
  • Do the results confirm findings in other studies? Should it be different? The paper would benefit from a connection of this interpretation with the results found in other countries

Author Response

Reply to Reviwer-2:

We would like to take this opportunity to thank the reviewer-2 of our manuscript for the crucial ideas and points which she/he has raised, below are the authors’ comments of each and every point raised by the reviewer-2.

  • The authors have considered the reviewer comment and shorten the abstract.
  • The research question defined in better way. See the yellow highlighted points in introduction part.
  • The introduction part reviewed and changed according to the suggestion of the revieer-2. Research topic, research gap and the connections with the Turkish economy is clearer now.
  • The authors have considered the reviewers comment on the precise of the research.
  • The reviewer suggested that section 3.2 may be need to eliminated because it looks redundant with section 4.1. But authors are not agreeing with reviewer at this point. Section 3.2 is not totally eliminated but the similar parts included both in the section 3.2 and 4.1 have been eliminated to avoid repetition.
  • The research results, limitations and directions for further research have been added to the conclusion part. The necessary comparison with the past studies has been done to emerge the importance and originality of the paper.
  • The manuscript was submitted for professional English Proof reading as per the reviewer’s comment.

Please see attachment as an revised version of the manuscript. 

Reviewer 3 Report

Review Report – “Nexus among competitively valued exchange rates, price level and growth performance in Turkey: Insight from the global value chain

 Strength 

The topic is timely and novel. However, there are technical problem and a lots of work required before the paper can be published in a good journal.

Weaknesses

  • The topic needs to be augmented as it not appropriate.
  • The abstract is poorly written. It needs to be re-motivated and reduced.
  • The present introduction does not state the problem addressed and the rationale for the study.
  • The authors cannot state the topic as contributions.
  • Theoretical review is part of literature review and should not appear in the same heading.
  • The opening paragraph of the literature review shows the authors do not understand the state of art of the subject. Also, the literature review is poorly written and not appropriate in this present version.
  • The appropriate channel through which exchange rate affects economic performance can be view from Mundell-Fleming hypothesis and not the approach used by the authors. Similarly, monetary policy variable is missing in the present model.
  • There the analysis is not sufficient to justify the conclusion of the authors. For example, the unit root is not correct. The results of the stationarity at level must be separated from the first difference. The ADF and PP is not sufficient. Therefore N.G perron unit should be used.
  • The author should implement structural break as the evidence of the stability test is insufficient. Bayer and Hanck cointegration test is require to complement the Bound test for cointegration. The authors claimed that vector error correction was used, whereas they Error correction regression has been implemented.

Decision 

Based on the bottleneck observed, I feel the paper cannot be published in the present version.

Author Response

Reply to Reviwer-3:

We would like to take this opportunity to thank the reviewer-3 of our manuscript for the crucial ideas and points which she/he has raised, below are the authors’ comments of each and every point raised by the reviewer-3.

  • The authors have considered the comments of the reviwer-3, the topic has been slightly modified.
  • The authors have considered the reviewer comment and shorten the abstract.
  • The authors have considered the reviewer comment and corrected the introduction part by (i) further explaining the relationship between the variables under study as well as highlighting the research gap. (ii) Authors have explained the contribution of the study to the literature. (iii) Variable selection and theoretical modelling have been explained. (iv) the aim of the study has been corrected to align it with the topic and the model used. See the points highlighted yellow in the introduction part.
  • The authors have considered the reviewer comment related with the Mundell-Fleming hypothesis and add a paragraph in section 2.1, theoretical framework part to not miss any particular point from the literature.
  • The authors have checked all of the test imposed and be sure they are correct. The stationarity test results are presented in the form of level and first difference to make it more compact and use the space efficiently. The authors think that the reviewer may mis specified the insufficiency of the ADF and PP and the efficiency of the NG Perron unit root test.
  • The authors have considered the suggestions of the reviewer and add the Bayer and Hanck cointegration test to complement the Bound test for cointegration.
  • The manuscript was submitted for professional English Proof reading as per the reviewer’s comment.

Please see attachment as an revised version of the manuscript. 

Round 2

Reviewer 1 Report

Dear authors, thank you very much for the changes made to the article. Please take into account publications on the risks associated with the activities of companies, and therefore also the economy. For example, Internet of Things and Other E-Solutions in Supply Chain Management May Generate Threats in the Energy Sector — The Quest for Preventive Measures. It is only an example of research work, but other articles can also be used and presented. I pointed out this issue because there are many risks associated with Global Value Chains worth presenting. I am writing about it because I am convinced that it will increase the cognitive value of this article.

Author Response

Please see the attached document as a revised version of the manuscript. 

Reviewer 2 Report

I think the new version of the paper reflects the suggestions.

Author Response

Ms. Ref. No/ID:

jrfm-1375036

Manuscript Title

The Nexus among Competitively Valued Exchange Rates, Price Level and Growth Performance in Turkey; New Insight from the Global Value Chain.

Subject

Response to Reviewers' comments

The authors would like to thank the referee for his objective and thorough review of our manuscript. We have addressed all the Reviewers’ comments in the following point-by-point response and prepared the revised manuscript. This report is prepared to highlight the reply to the comments, to make it clear for the reviewer. We indicated our reply in different colors to differentiate between the comments and the authors’ reply as Black and Blue, respectively.

All changes made to accommodate the Reviewers' comments are highlighted by red color or tracked changes in the revised manuscript.

The authors hope that you find the revised paper in the level of publication in Journal of Risk and Financial Management.

.

Thank you for your patience and support in this process.

October. 2021

Reviewer 3 Report

Dear Authors(s), 

I review the revised version of the manuscript but it appears additional work is require before it can be publishable. 

  1. The abstract is weak. The author(s) need to remove the interpretation of the error correction term and state the result and policy implication/recommendation.
  2. The introduction is confusing. The author(s) need to arrange their thought and spell out the contribution of their study both theoretically and empirically. 
  3. Figure 2 indicating the normality test should be removed and the figure should be reported in Table 7. 
  4. The author(s) should compare their study with existing one and provide suggestion for future studies.

Author Response

Ms. Ref. No/ID:

jrfm-1375036

Manuscript Title

The Nexus among Competitively Valued Exchange Rates, Price Level and Growth Performance in Turkey; New Insight from the Global Value Chain.

Subject

Response to Reviewers' comments

The authors would like to thank the referee for his objective and thorough review of our manuscript. We have addressed all the Reviewers’ comments in the following point-by-point response and prepared the revised manuscript. This report is prepared to highlight the reply to the comments, to make it clear for the reviewer. We indicated our reply in different colors to differentiate between the comments and the authors’ reply as Black and Blue, respectively.

All changes made to accommodate the Reviewers' comments are highlighted by red color or tracked changes in the revised manuscript.

The authors hope that you find the revised paper in the level of publication in Journal of Risk and Financial Management.

Thank you for your patience and support in this process.

October. 2021.

Round 3

Reviewer 3 Report

Dear Authors, 

Thank you for sending your revised paper. while the version appears to have improved, the equations needs to be corrected. 

Similarly, the author discusses more on technology and the no technical or technology variable is included. 

The empirical results need to be discussed and compared with existing literature. 

The policy implications must be spell out in the conclusion and limitation of the study should be included. 

I also think the lag value to each variable in the Table 5 should be discuss and possible reason for the results should be explained. 

Reviewer

Round 4

Reviewer 3 Report

Dear Author(s), 

The equations is still not corrected. 

For example, equation (s) used Ut as the stochastic term, while the rest of the equations is Et. 

Similarly, the policy implication is yet to be spell out. 

I think the author must address the comments before the paper can be publishable. 

Regards, 

Reviewer

Round 5

Reviewer 3 Report

Dear Author(s), 

I think the paper has greatly improved and satifactory. 

Regards, 

Reviewer